# The Effect of *Dekkera bruxellensis* Concentration and Inoculation Time on Biochemical Changes and Cellulose Biosynthesis by *Komagataeibacter intermedius*

**DOI:** 10.3390/jof8111206

**Published:** 2022-11-15

**Authors:** Putu Virgina Partha Devanthi, Ferren Pratama, Katherine Kho, Mohammad J. Taherzadeh, Solmaz Aslanzadeh

**Affiliations:** 1Indonesia International Institute for Life Sciences, Pulomas Barat Kavling 88, Jakarta 13210, Indonesia; 2Swedish Centre for Resource Recovery, University of Borås, 50190 Borås, Sweden

**Keywords:** co-culture, *Komagataeibacter intermedius*, *Dekkera bruxellensis*, bacterial cellulose, kombucha

## Abstract

Bacterial Cellulose (BC) is a biopolymer with numerous applications. The growth of BC-producing bacteria, *Komagataeibacter intermedius*, could be stimulated by *Dekkera bruxellensis*, however, the effect on BC yield needs further investigation. This study investigates BC production and biochemical changes in the *K. intermedius-D. bruxellensis* co-culture system. *D. bruxellensis* was introduced at various concentrations (10^3^ and 10^6^ CFU/mL) and inoculation times (days 0 and 3). BC yield was ~24% lower when *D. bruxellensis* was added at 10^3^ CFU/mL compared to *K. intermedius* alone (0.63 ± 0.11 g/L). The lowest BC yield was observed when 10^3^ CFU/mL yeast was added on day 0, which could be compromised by higher gluconic acid production (10.08 g/L). In contrast, BC yields increased by ~88% when 10^6^ CFU/mL *D. bruxellensis* was added, regardless of inoculation time. High BC yield might correlate with faster sugar consumption or increased ethanol production when 10^6^ CFU/mL *D. bruxellensis* was added on day 0. These results suggest that cell concentration and inoculation time have crucial impacts on species interactions in the co-culture system and product yield.

## 1. Introduction

Bacterial cellulose (BC) refers to the biomaterial of glucose monomers produced by bacteria [1]. Cellulose is mainly obtained from plants. However, BC was reported to possess attractive properties, including higher purity, crystallinity, tensile strength, water-holding capacity, thermal stability and malleability [2,3]. Hence, BC is used for different applications, such as wound healing, drug delivery, as potential electric capacitors, and as a food additive [1,2].

The bacteria genus Komagataeibacter is well known for its ability to produce BC [4]. This genus is also abundant during the production of *kombucha*, a fermented sweetened tea, where they also form BC [5]. However, *kombucha* is not produced by the bacteria alone but with the co-culture of various yeasts species [5]. It is believed that the yeasts break down sucrose to glucose and fructose, while the bacteria utilize it to form BC and carboxylic acids [4,6]. In addition, yeasts can also produce ethanol, which was reported to increase BC production and reduce the presence of non-BC producing bacteria [7,8]. Lastly, the bacteria can convert the ethanol to acetic acid, which has also been found to increase BC yield [9]. Due to the ability to increase the amount of reducing sugars and advantageous metabolites, it is, therefore, likely that the presence of yeasts can potentially promote BC production. 

*Komagataeibacter intermedius* is one of the BC-producing bacteria found in *kombucha* and was isolated in a previous study [10]. *K. intermedius* was reported to produce more BC than *Komagataeibacter xylinus*, the model BC producer [11,12,13]. *K. intermedius* also produced a high BC yield in sugarcane molasses, a cheap alternative to current commercial media for BC production [10,14]. Additionally, it was also found that the survival rate of *K. intermedius* increased in the presence of a *kombucha* isolated yeast, *D. bruxellensis*; however, BC production in a co-culture required improvement [10]. This result indicates the synergism of the yeast and the bacteria, and further optimization of yeast inoculation is required to achieve a higher BC yield. 

Besides the effects of using a co-culture, the concentration of the yeast and the time of addition have also been found to influence not only the co-culture interactions but also the end product’s characteristics. In *kombucha,* for example, it was reported that a lower yeast concentration increased the amount of glucuronic acid and gluconic acid produced by *Gluconacetobacter intermedius* [15,16]. Furthermore, adding yeast sequentially has also been reported to impact the properties of other kinds of fermented products. For instance, in sour beer, adding yeast sequentially resulted in a higher yield of lactic acid [17]. In soy sauce, sequentially adding of *Zygosaccharomyces rouxii* increased the alcohol and ketone content during soy sauce fermentation and prolonged the bacteria’s survival rate [18]. Yet, there has not been any research on the effect of different concentrations of yeast or sequential addition with an emphasis on BC production. 

Therefore, this study aimed to investigate the effect of different concentrations of *D. bruxellensis* added simultaneously and sequentially to the cultivation medium inoculated with *K. intermedius*. The effect of yeast was analyzed by observing the microbial growth, the BC yield, and metabolites, including ethanol, free amino nitrogen, gluconic and glucuronic acid concentration during the cultivation. 

## 2. Materials and Methods

### 2.1. Materials 

*K. intermedius* and *D. bruxellensis* culture used in this study were previously isolated by Devanthi et al. [10] and were stored in 1.5 mL of 20% glycerol solution at −80 °C until use. PT Andalan Furnindo, Indonesia, supplied molasses with brown/black color, containing 77.6% Brix, 46.9% sucrose, 50.8% inverted sugar, 1.4% specific gravity and 60.8% purity. Pure caffeine (PureBulk, Roseburg, OR, USA) was purchased from the local market in Jakarta. The pH of molasses was maintained by adding an acetate buffer prepared using acetic acid (Merck, Darmstadt, Germany) and sodium acetate (Merck, Darmstadt, Germany). Microbiological growth media used were Potato Dextrose Broth (PDB) and Potato Dextrose Agar (PDA) (HiMedia, Mumbai, India) and Hestrin Schramm (HS) comprised of 20 g/L glucose, 5 g/L peptone, 5 g/L yeast extract, 2.7 g/L Na_2_HPO_4_, and 1.15 g/L citric acid. For cell enumeration, the bacteria and yeast growth were controlled by supplementing the agar media with acetic acid (Merck, Darmstadt, Germany) and NaCl (Merck, Darmstadt, Germany), respectively.

### 2.2. Methods

#### 2.2.1. Media and Culture Preparation 

*K. intermedius* and *D. bruxellensis* cultures were prepared by culturing the frozen stock on HS agar and PDA, respectively, for 5 days at 30 °C. Then, a loopful of each microbe was transferred to 100 mL of the respective broth media and incubated for 24 h at 30 °C under static conditions. *K. intermedius* cell concentration was adjusted to ~10^6^ CFU/mL. In order to evaluate the effect of *D. bruxellensis* concentration on BC production, the cell concentrations were adjusted to ~10^6^ CFU/mL and ~10^3^ CFU/mL. The cell concentration for both *K. intermedius* and *D. bruxellensis* was adjusted based on the standard curve (OD vs. CFU/mL) generated prior to the experiment. Cultivation media was prepared by dissolving 150 g/L of sugarcane molasses and 500 mg/L of caffeine into acetate buffer (pH 4.75, 0.2 M). 

#### 2.2.2. BC Production Set Up 

BC production was carried out using 6-well plates. Each well consisted of 9 mL cultivation media inoculated with 1 ml of inoculum. As shown in Table 1, the inoculum contained *K. intermedius* and *D. bruxellensis* either in monoculture (B_-control_ and Y_-control_, respectively) or co-cultures (BY_H0,_ BY_L0,_ BY_H3,_ and BY_L3_). The concentration of *K. intermedius* was fixed at ~10^6^ CFU/mL in all samples. Meanwhile, *D. bruxellensis* concentration was varied; ~10^6^ CFU/mL in Y_-Control_, BY_H0_ and BY_H3_ and ~10^3^ CFU/mL in BY_L0_ and BY_L3_. *D. bruxellensis* was added simultaneously on day 0 in BY_H0_ and BY_L0_ or sequentially on day 3 in BY_H3_ and BY_L3_. All samples were incubated in a static condition at 30 °C for 14 days. Samples were taken on days 0, 3, 7 and 14. For the sequential culture, extra samples were also collected on day 5. All experiments were conducted in triplicate and were subjected to bacteria and yeast enumeration, as well as pH, sugars, acids, ethanol, and free amino nitrogen analyses.

#### 2.2.3. *K. intermedius* and *D. bruxellensis* Cell Enumeration

*K. intermedius* and *D. bruxellensis* enumeration was carried out by collecting 0.1 mL of the cultivation broth on each sampling day. The samples were then serially diluted up to 10^−8^ using 0.85% NaCl solution and were plated on HS agar supplemented with 1% *v*/*v* acetic acid and PDA with 2% *w*/*v* NaCl for *K. intermedius* and *D. bruxellensis*, respectively. A previous study demonstrated that the addition of 2% *w*/*v* NaCl or 1% *v*/*v* acetic acid was able to inhibit the growth of either *K. intermedius* or *D. bruxellensis*, respectively, which allowed the observation of each microbe from the co-culture [19]. The colonies were enumerated after 3 days of incubation at 30 °C.

Specific growth rate μ (h^−1^) were calculated by Equation (1) [20]: (1)$$\mu =\frac{\ln\left(X_{0}-X_{t}\right)}{t-t_{0}}$$
where

*X_t_* and *X*_0_ are the microbial population (CFU mL^−1^) at t and initial time, respectively;

*t* and *t*_0_ are the t and initial time when the sample is measured, respectively;

μ is specific growth rate (1 h^−1^).

#### 2.2.4. Biochemical Analysis 

Changes in pH during the cultivation were monitored using a pH meter (ST300, OHAUS, Parsippany, NJ, USA). Prior to biochemical analysis, samples were centrifuged at 1000× *g* for 10 min at room temperature. Then, the supernatant was collected into a new tube, and the process was repeated until no pellet was visually observed. Sugars (glucose and fructose) were measured using the K-SUFRG kit, ethanol was measured using K-ETOH kit, and acids (gluconic and glucuronic acids) were measured using K-GATE and K-URONIC kits (Megazyme, International Ireland Ltd., Bray, Ireland), according to the manufacturer’s instructions. 

Free amino nitrogen (FAN) concentration was measured using the ninhydrin analysis method [21]. The ninhydrin color reagent consisted of 0.3 g fructose, 6 g KH_2_PO_4_, 10 g Na_2_HPO_4_ and 0.5 g ninhydrin dissolved in 100 mL of distilled water. A solution mixture was prepared by dissolving 2 g of potassium iodide into 600 mL of distilled water and 400 mL of 96% ethanol. The samples were first diluted 50 times using distilled water. Then, 2 mL of the diluted sample was mixed with 1 mL of the ninhydrin color reagent. The mixture was boiled for 16 minutes and left to cool down to room temperature using an ice bath for 20 min. Afterwards, 5 mL of the previously prepared solution mixture was added to each sample, and the absorbance was measured at 570 nm. 

#### 2.2.5. BC Yield Measurement

After 14 days of incubation, BC formed on the surface of the medium were collected and rinsed in distilled water until no excess media remained on the pellicle. The BC samples were then treated using 1 M NaOH at 80 °C for at least an hour to remove the remaining microbes and leftover broth. The treated samples were then rinsed in distilled water to remove the NaOH before being oven dried at 60 °C overnight until a stable weight was achieved. The dried BC weight was measured using an analytical balance, and the BC yield was reported as gram dry weight per liter of culture volume (g/L). 

#### 2.2.6. Statistical Analysis

The data were analyzed using R studio program [22] for one-way analysis of variance (ANOVA). One-way ANOVA followed by Tukey HSD test were carried out to investigate if the presence of yeast in each sample had any effect on the final BC yield and glucuronic acid concentration. A two-way ANOVA was also done on BC yield to investigate the interaction effects between the yeast concentration and inoculation time. The effect was considered statistically significant if the p-value was less than or equal to the selected significance level (*p*-values < 0.05).

## 3. Results and Discussion

### 3.1. Effect of D. bruxellensis Concentration and Inoculation Time on Bacteria-Yeast Interactions

In *kombucha*, yeasts are hypothesized to support bacteria by breaking down sucrose or providing ethanol as an additional carbon source [7]. Bacteria-yeast co-cultures in kefir and sourdough bread were also known to support each other by producing other essential nutrients, such as vitamin B6 and amino acids [23,24]. However, other studies reported that depending on the initial cell ratio, a microbe can outcompete the other by depleting the available nutrients or producing inhibitory compounds [25,26]. Furthermore, adding yeast sequentially can also impact bacterial fermentation, which has been observed in sour beer and soy sauce [17,18]. Therefore, *K. intermedius* productivity may also be affected by the cell ratio and inoculation time of *D. bruxellensis*.

As shown in Figure 1, *K. intermedius* in all samples had a similar specific growth rate (~0.374 h^−1^), indicating that the *D. bruxellensis* may not be antagonistic. Furthermore, *K. intermedius* in samples with *D. bruxellensis* added sequentially (BY_H3_ and BY_L3_) were also able to maintain a high concentration up to day 5 of the experiment. However, similar to B_-control_, *K. intermedius* in BY_H3_ and BY_L3_ also had no observable growth by the end of cultivation. On the other hand, samples with simultaneous addition of *D. bruxellensis* had a high concentration of *K. intermedius* up to day 7 and had observed bacterial growth until the end of cultivation, with 6.14 ± 0.28 log CFU/mL in BY_H0_ and 3.01 ± 0.54 log CFU/mL in BY_L0_. The simultaneous addition of *D. bruxellensis* may have provided the *K. intermedius* with more resources, such as amino acids and ethanol, which allowed *K. intermedius* to survive for an extended time [6,8].

*D. bruxellensis* population in yeast mono (Y_-control_) and co-cultures was also monitored. The initial counts of *D. bruxellensis* were lower by 1 log CFU/mL than the adjusted goal, which could be attributed to the accuracy of OD to log CFU/mL conversion, resulting from variations in yeast’s physiological state and ability to grow on the agar. Then, adding the inoculum into the fermentation media would further dilute the cell concentration.

*D. bruxellensis* in Y_-control_ and BY_H0_ were observed to grow at a similar rate reaching a final concentration of ~8 log CFU/mL by day 14 (Figure 2). While *D. bruxellensis* in BY_L0_ experienced a significantly slower initial growth, it grew to the same concentration as Y_-control_ and BY_H0_ by day 7. Samples with the sequential addition of *D. bruxellensis* grew even slower. BY_H3_ experienced an initial decrease before fluctuating at approximately 5 log CFU/mL until the end of cultivation, while no growth of *D. bruxellensis* could be observed throughout incubation in BY_L3_. This suggests that adding *D. bruxellensis* at the later stage (on day 3), especially at a lower concentration, could be unfavorable for yeast. By that time, *K. intermedius* almost enters the stationary phase (Figure 1), leaving less resources available and more harmful inhibitory compounds, such as acetaldehyde [27,28], creating a hostile environment for *D. bruxellensis*.

### 3.2. Effect of D. bruxellensis Concentration and Inoculation Time on pH Changes 

*Komagataeibacter* are well known for producing gluconic, glucuronic, and acetic acids [5]. *D. bruxellensis* is also known to produce acetic acid [29]. The accumulation of acids during cultivation may impact the pH, affecting their growth and eventually the BC production. BC production can be inhibited when the pH is outside of the bacteria’s optimum range [30]. Therefore, the pH of the media was monitored throughout the incubation. It was found that the pH of all samples increased to a final pH of at least 8, which was consistent with the previous work [10]. Such an increasing pH trend was rarely observed in acetic acid bacteria fermentation, which is more likely to cause pH reduction due to organic acid production [5,31]. Previous work reported that the pH increase was associated with the presence of acetate buffer in the medium [10,32]. Furthermore, ammonia production by *K. intermedius* might have also contributed to the pH increase, which is further discussed in Section 3.6. The ability of *K. intermedius* to produce BC at a pH of 9 has been previously reported [10,11]. Although the optimal pH for BC production is typically slightly acidic, different strains have also been found to prefer more alkaline conditions [30,33].

Even though the pH in all samples increased, the rate of the increase seemed to be dependent on the concentration and inoculation time of *D. bruxellensis* (Figure 3). In B_-control_, the pH started to increase from day 3 to a final pH of 9.31 ± 0.02. The pH in samples with simultaneous addition of *D. bruxellensis* also experienced an increase from day 3, but at a slower rate. BY_H0_ obtained the lowest final pH among all samples (7.96 ± 0.31), followed by BY_L0_ (8.75 ± 0.1). In samples with the sequential addition of *D. bruxellensis*, the pH only increased after day 5 but resulted in a similar final pH as B_-control_. 

### 3.3. Effect of D. bruxellensis Concentration and Inoculation Time on Sugar Concentration during Co-Culture with K. intermedius 

Bacteria typically produce BC and other acids from glucose; hence, the production of both relies on the amount of available sugars. Yeasts may provide more glucose by breaking down available sucrose, but they can also potentially outcompete bacteria and consume the reducing sugars instead [7]. Therefore, the amount of reducing sugars was measured throughout the incubation to detect any competition for sugar when *D. bruxellensis* was added at a lower concentration or added sequentially. 

Figure 4 shows that *K. intermedius* in B_-control_ prefers glucose, with 90% consumed by day 7 of the incubation, while fructose concentration remains constant until day 14 (Figure 4A,B). Similarly, BY_H0_ experienced the same trend for glucose consumption with approximately 90% of the sugar consumed by day 7. Meanwhile, the glucose consumption rate in BY_L0_ was slower, with the same amount of sugar consumed by day 14. It was also found that BY_H0_ and BY_L0_ utilized fructose as well. When *D. bruxellensis* was added sequentially (BY_H3_ and BY_L3_), an increase in glucose concentration was observed by day 5. By the end of the incubation, BY_H3_ used up roughly 82% of the glucose, while BY_L3_ consumed the least amount (~70%). The fructose in both samples showed a slight increase by the end of incubation, which indicates that the *D. bruxellensis* may also only utilize fructose when glucose is unavailable. 

Other studies have reported that *K. intermedius* could utilize fructose, although they still primarily prefer glucose [14,34]. It is also noted that different strains of *Komagataeibacter* may prefer different kinds of sugars [35]. In another study it was demonstrated that the co-culture consume the total sugars at a higher rate than bacterial monoculture [10]. Therefore, total sugars may be used more quickly especially when the yeast is added simultaneously [36].

### 3.4. Effect of D. bruxellensis Concentration and Inoculation Time on Gluconic and Glucuronic Acid Production by K. intermedius

Gluconic and glucuronic acid are commonly found during *kombucha* fermentation and result from the oxidation of glucose’s 1st and 6th carbon [5,37]. In *kombucha*, the yeast has been observed to stimulate acid production by acetic acid bacteria, either by providing more readily available carbon sources or by producing other beneficial nutrients [5,6]. However, increased acid production may impact BC production by using more glucose or acidifying the environment [30]. Hence, gluconic and glucuronic acid production were detected in *K. intermedius* monoculture and co-cultures. 

As shown in Figure 5, the gluconic acid concentration in all samples peaked by day 5 and then declined toward the end of the incubation period. The production followed by consumption of gluconic acid has been noted in other studies in *Acetobacter xylinum* NUST4.2 and *Komagataeibacter hansenii* [31,38]. While B_-control_ peaked by day 5 at 8.45 ± 0.45 g/L, both co-cultures with simultaneous addition of yeast peaked earlier at day 3. However, BY_H0_ peaked with 16% less gluconic acid, while BY_L0_ produced 20% more gluconic acid than B_-control_. Similar results were also found in another study where a lower yeast ratio in *Starmerella davenportii* (yeast)-*Gluconacetobacter intermedius* (bacteria) co-culture produced thrice the amount of gluconic acid compared to bacteria monoculture due to more glucose produced from sucrose breakdown [6,16]. However, samples with *D. bruxellensis* added sequentially had a lower gluconic acid production rate and yield than B_-control_ but more sugars remaining (Figure 4). The higher production rate in BY_H0_ and BY_L0_ could be due to *D. bruxellensis* being more competitive forcing *K. intermedius* to produce acids or perhaps the metabolites produced by *D. bruxellensis* could stimulate acid production, as its presence has been found to correlate with higher acid production in *kombucha* [6,39].

*K. intermedius* in monoculture (B_-control_) produced 22.9 ± 1.39 mg/L of glucuronic acid by the end of incubation. Most co-cultures, however, obtained a higher concentration (Figure 6). Regardless of inoculation time, samples with a high concentration of *D. bruxellensis* produced approximately 30 mg/L. On the other hand, adding a low concentration of *D. bruxellensis* on day 0 (BY_L0_) gave the highest concentration (35.5 ± 0.29 mg/L), while adding the yeast on day 3 (BY_L3_) showed the lowest concentration (14.5 ± 2.05 mg/L). Previous studies on co-culture have shown that the presence of *D. bruxellensis* in general is able to promote glucuronic acid production by *Gluconacetobacter intermedius* [15]. The higher yield could either be due to *D. bruxellensis* providing more glucose from sucrose breakdown or by producing acetic acid, which can inhibit glycolysis in *G. intermedius* and therefore stimulate glucuronate synthesis [15]. Furthermore, the optimal ratio of *D. bruxellensis* to *G. intermedius* was 4:6, similar to the inoculum condition of BY_L0_ [15]. 

### 3.5. Effect of D. bruxellensis Concentration and Inoculation Time on Ethanol Production 

Ethanol is known to improve BC synthesis as an additional energy source allowing glucose to be used mainly for BC production only [9]. Ethanol is also an alternative carbon source used by acetic acid bacteria [24]. In previous studies, *D. bruxellensis* isolated from *kombucha* has been shown to produce a high amount of ethanol [8]. Therefore, the ethanol production by *D. bruxellensis* during co-culture *with K. intermedius* was observed. The results confirm the previous studies as the ethanol production increased over time and peaked by day 14 (Figure 7). By day 14, Y_-Control_ showed the highest final concentration with 6952 ± 923 mg/L (data not shown) followed by BY_H0_ with 1235 ± 256 mg/L, while BY_L0_ showed considerably lower concentration and produced only 105 ± 9 mg/L by the end of the incubation. Samples with *D. bruxellensis* added sequentially resulted in a lower final ethanol concentration, producing 19 ± 1.4 mg/L in BY_H3_ and a negligible amount in BY_L3_, which corresponds to the yeast’s undetectable growth (Figure 2). This finding indicates that inoculating *D. bruxellensis* simultaneously at 10^6^ CFU/mL enables yeast to reach a high population more rapidly, which is crucial for increased ethanol production. 

### 3.6. Effect of D. bruxellensis Concentration and Inoculation Time on Free Amino Nitrogen (FAN) Concentration

The absence of a nitrogen source is known to result in 30% less BC yield, while an overabundance of nitrogen may instead only promote *Komagataeibacter* growth [40]. A nitrogen source is also crucial for *D. bruxellensis* growth, and the yeast has also been known to produce FAN, which is theorized to stimulate bacterial activity [6]. Therefore, the FAN concentration was measured in all samples during cultivation to investigate how the microbes interact (Figure 8). It was found that FAN in *K. intermedius* monoculture (B_-control_) only fluctuated slightly, with less than 5% being used, resulting in a final concentration of 96.78 ± 6.1 mg/L. This indicates that *K. intermedius* only needs a certain amount of nitrogen. Several studies have also reported that nitrogen consumption may vary depending on the *Komagataeibacter* strain and available sources [40,41]. A consistent decline was observed when a low concentration of *D. bruxellensis* was added on day 0 (BY_L0_), with only 16% of the FAN remaining by the end of the incubation (15.93 ± 2.6 mg/L). This could be because the FAN has been used for the yeast in BY_L0_ to grow rapidly (Figure 2). When a low concentration of *D. bruxellensis* was added on day 3 (BY_L3_), the FAN concentration followed a similar trend as B_-control_, with a slightly higher final concentration (115.6 ± 4.3 mg/L). The higher final concentration in BY_L3_ may be due to autolysis of *D. bruxellensis* as it could not compete with *K. intermedius* [6,42]. 

In both BY_H0_ and BY_H3_, an initial increase in FAN can be observed, with both samples peaking above 140 mg/L on day 3 and day 5, respectively. However, the FAN concentration then declined to approximately 75.24 ± 1.84 mg/L in BY_H3_ by day 14, while in BY_H0_, the FAN concentration decreased rapidly up to day 7 and then increased again toward the end of incubation from 36.61 ± 1.42 mg/L to 107.41 ± 11.64 mg/L. These results are similar to the findings of Tran et al. [6], where an equal amount of *Komagataeibacter saccharivorans* and *D. bruxellensis* bacteria-yeast co-culture was found to have a higher final FAN concentration compared to pure bacterial culture. However, it should be noted that the ninhydrin method can be used to detect small proteins and ammonia as well [43]. Yeasts in *kombucha* are known to produce amino acids from the available nitrogen in tea. *D. bruxellensis* is also particularly known for its ability to produce γ-Aminobutyric acid (GABA) [44]. Finally, acetic acid bacteria are known to produce ammonia to survive acidic conditions [45]. Therefore, the final FAN concentration in this study may indicate not only amino acid consumption but also ammonia production, which could explain the pH increase toward the end of incubation (Figure 3).

### 3.7. Effect of D. bruxellensis Concentration and Inoculation Time on BC Production by K. intermedius

Varying yeast concentrations and inoculation time may affect how bacteria and yeast interact, eventually impacting BC production. This study found that adding *D. bruxellensis* at a high concentration (10^6^ CFU/mL), regardless of inoculation time, could enhance BC yield by 74–102% (Figure 9). B_-control_ could only produce 0.63 ± 0.11 g/L BC after 14 days, while the yield increased to 1.09 ± 0.02 g/L and 1.27 ± 0.33 g/L when a high concentration of *D. bruxellensis* was simultaneously (BY_H0_) and sequentially (BY_H3_)added, respectively. On the other hand, adding *D. bruxellensis* at a low concentration (10^3^ CFU/mL), regardless of the inoculation time, decreased the BC yield less than that of the bacteria alone. The lowest BC yield was observed in the co-culture with a low concentration of *D. bruxellensis* added simultaneously (BY_L0_), which produced 0.44 ± 0.012 g/L, a 30% lower yield than B_-control,_ while sequential addition *D. bruxellensis* (BY_L3_) had 18% lower BC yield compared to B_-control_ (0.52 ± 0.017 g/L). The BC yield is noticeably lower in this study compared to our previous study [10], potentially due to NaOH treatment for BC purification. Other studies have found that NaOH treatment of BC at high temperatures resulted in a lower BC mass due to interactions with the BC structure [46,47]. Additionally, the BC mass could further decrease due to NaOH treatment being more intensive to completely remove the dark colored sugarcane molasses-based media. 

In this study, it was observed that a lower concentration of *D. bruxellensis* (BY_L0_) gave a lower BC yield and more gluconic acid (10.08 ± 0.12 g/L), while BY_H3_ produced less gluconic acid (4.64 ± 0.19 g/L) and gave a higher BC yield. According to a study by Gilbert et al. [48], adding a lower concentration of *S. cerevisiae* to *Komagataeibacter rhaeticus* could increase the BC yield due to less competition for glucose. Perhaps the sequential addition of *D. bruxellensis* (BY_H3_) was more similar to this condition since more glucose was available for the bacteria (Figure 4A). In addition to being competitive for glucose, BY_L0_ also promoted gluconic acid production, which may have caused less BC to be produced [49]. On the other hand, the high BC yield observed in BY_H0_ could be due to the *D. bruxellensis* supporting the *K. intermedius* growth for a longer period (Figure 1), either by ethanol or FAN production (Figure 7 and Figure 8) [6,48]. 

The significance of the *D. bruxellensis* concentration and inoculation time was calculated with the General Linear Model at a 95% confidence interval. The result shows that the *D. bruxellensis* concentration (*p*-value 0.01) and the time of inoculation (*p*-value 0.031) are significant on the final BC yield. However, the interrelation between *D. bruxellensis* ratio and the inoculation time (*p*-value 0.21) at a 5% confidence level was not significant on the final BC yield. 

## 4. Conclusions 

This study aimed to investigate the effect of *D. bruxellensis* concentrations (10^3^ and 10^6^ CFU/mL) and inoculation time (days 0 and 3) to promote BC synthesis by *K. intermedius*. The results show that regardless of inoculation time, a lower concentration of *D. bruxellensis* (10^3^ CFU/mL) reduces the BC yield by 18–30% (0.44–0.52 g/L) compared to that of *K. intermedius* in monoculture (0.63 ± 0.11 g/L). On the other hand, adding a higher concentration of *D. bruxellensis* (10^6^ CFU/mL), either simultaneously or sequentially, could increase the BC yields by 74–102% (1.09–1.27 g/L). Adding a higher *D. bruxellensis* concentration at the start of incubation could improve the survival of both species and stimulate metabolic reactions favorable for BC production, such as increased sugar consumption and higher ethanol and FAN production. In contrast, lower *D. bruxellensis* concentration promotes gluconic and glucuronic acid production, resulting in lower BC yields. These results show that it is feasible to regulate the BC production of *K. intermedius* in co-culture by controlling the inoculum proportion and time of addition of *D. bruxellensis*. Prior to scaling up the co-culture, more studies can be done on the metabolomics of the co-cultures in response to the varying ratios and inoculation time of *D. bruxellensis*. After identifying the role of each metabolite produced by *D. bruxellensis*, the metabolites responsible for stimulating BC production can be further optimized, while metabolites that stimulate acid production can be limited. 

## Figures and Tables

**Figure 1 jof-08-01206-f001:**
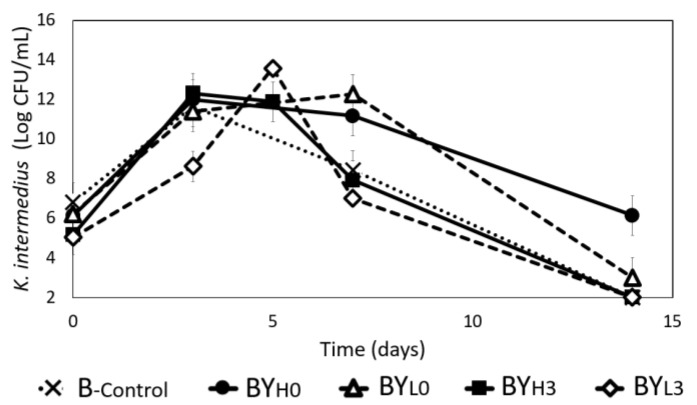
*K. intermedius* growth during 14 days of cultivation in mono (B_-control_) and co-cultures with *D. bruxellensis* at 30 °C. *D. bruxellensis* was added simultaneously at concentrations of 10^6^ CFU/mL (BY_H0_) and 10^3^ CFU/mL (BY_L0_) and sequentially at 10^6^ CFU/mL (BY_H3_) and 10^3^ CFU/mL (BY_L3_).

**Figure 2 jof-08-01206-f002:**
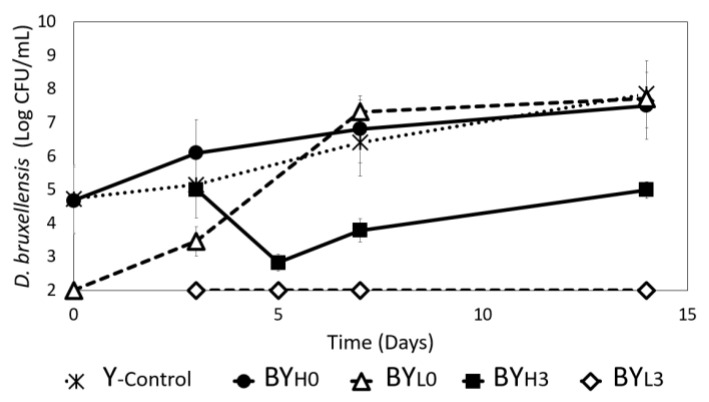
*D. bruxellensis* growth during 14 days cultivation in mono (Y_-control_) and co-cultures with *K. intermedius* at 30 °C. *D. bruxellensis* was added simultaneously at concentrations of 10^6^ CFU/mL (BY_H0_) and 10^3^ CFU/mL (BY_L0_) and sequentially at 10^6^ CFU/mL (BY_H3_) and 10^3^ CFU/mL (BY_L3_).

**Figure 3 jof-08-01206-f003:**
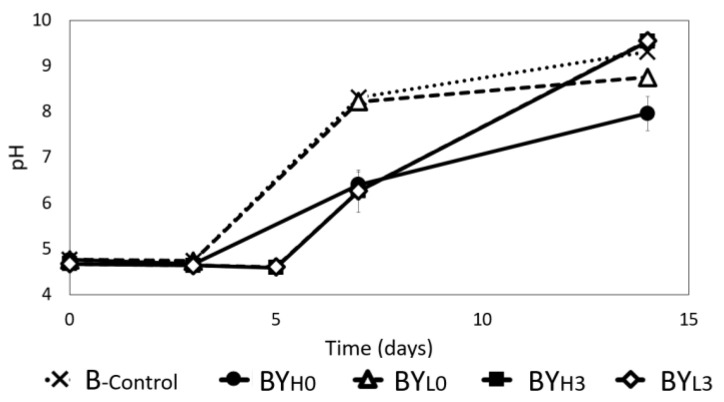
Changes in pH during 14 days of cultivation using *K. intermedius* mono-(B_-control_) and co-cultures with *D. bruxellensis* at 30 °C. *D. bruxellensis* was added simultaneously at concentrations of 10^6^ CFU/mL (BY_H0_) and 10^3^ CFU/mL (BY_L0_) and sequentially at 10^6^ CFU/mL (BY_H3_) and 10^3^ CFU/mL (BY_L3_).

**Figure 4 jof-08-01206-f004:**
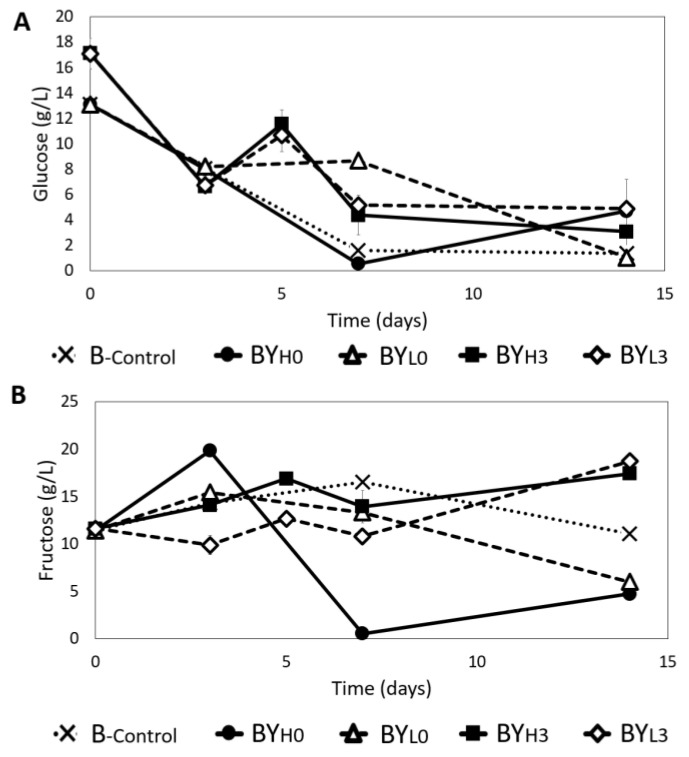
Changes in (**A**) glucose and (**B**) fructose concentration during 14 days of cultivation using *K. intermedius* mono-(B_-control_) and co-cultures with *D. bruxellensis* at 30 °C. *D. bruxellensis* was added simultaneously at concentrations of 10^6^ CFU/mL (BY_H0_) and 10^3^ CFU/mL (BY_L0_) and sequentially at 10^6^ CFU/mL (BY_H3_) and 10^3^ CFU/mL (BY_L3_).

**Figure 5 jof-08-01206-f005:**
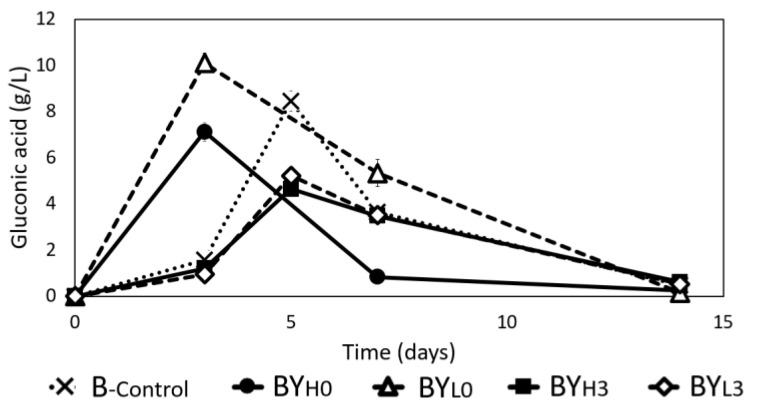
Changes in gluconic acid during 14 days of cultivation using *K. intermedius* mono-(B_-control_) and co-cultures with *D. bruxellensis* at 30 °C. *D. bruxellensis* was added simultaneously at concentrations of 10^6^ CFU/mL (BY_H0_) and 10^3^ CFU/mL (BY_L0_) and sequentially at 10^6^ CFU/mL (BY_H3_) and 10^3^ CFU/mL (BY_L3_).

**Figure 6 jof-08-01206-f006:**
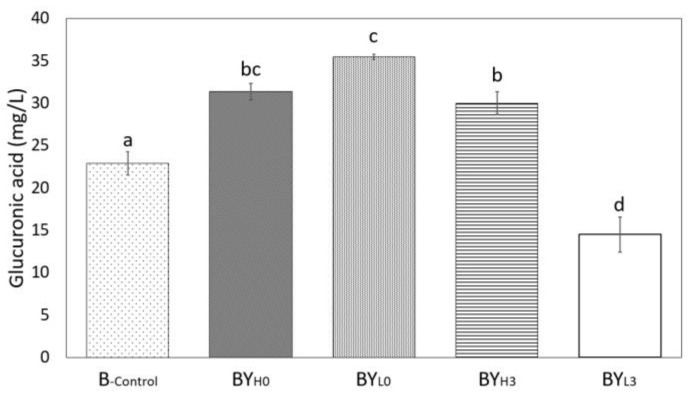
Glucuronic acid concentration after 14 days of cultivation using *K. intermedius* mono-(B_-control_) and co-cultures with *D. bruxellensis* at 30 °C. *D. bruxellensis* was added simultaneously at concentrations of 10^6^ CFU/mL (BY_H0_) and 10^3^ CFU/mL (BY_L0_) and sequentially at 10^6^ CFU/mL (BY_H3_) and 10^3^ CFU/mL (BY_L3_). Means with different letters are significantly different (*p* < 0.05).

**Figure 7 jof-08-01206-f007:**
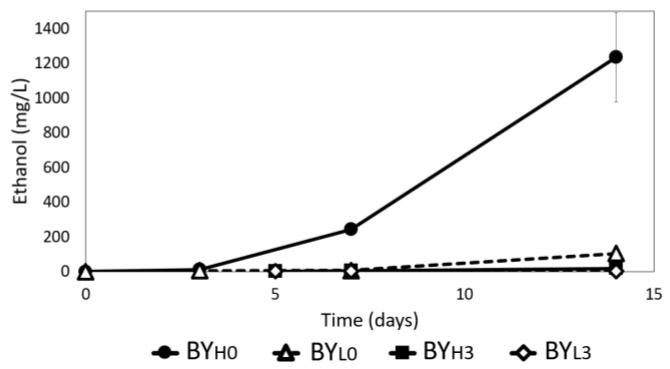
Ethanol production by *D. bruxellensis* during 14 days of co-culture with *K. intermedius* at 30 °C. *D. bruxellensis* was added simultaneously at concentrations of 10^6^ CFU/mL (BY_H0_) and 10^3^ CFU/mL (BY_L0_) and sequentially at 10^6^ CFU/mL (BY_H3_) and 10^3^ CFU/mL (BY_L3_).

**Figure 8 jof-08-01206-f008:**
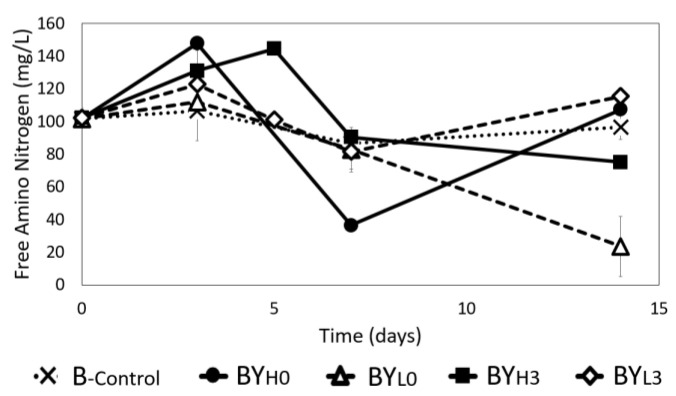
Changes in free amino nitrogen (FAN) concentration during 14 days of cultivation using *K. intermedius* mono-(B_-control_) and co-cultures with *D. bruxellensis* at 30 °C. *D. bruxellensis* was added simultaneously at concentrations of 10^6^ CFU/mL (BY_H0_) and 10^3^ CFU/mL (BY_L0_) and sequentially at 10^6^ CFU/mL (BY_H3_) and 10^3^ CFU/mL (BY_L3_).

**Figure 9 jof-08-01206-f009:**
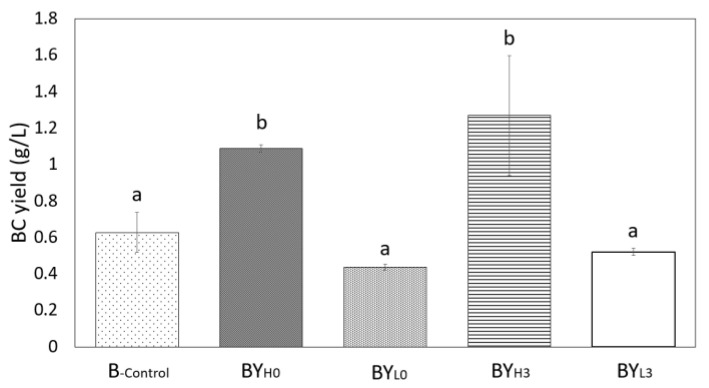
Effect of *D. bruxellensis* concentration and inoculation time on BC final yield by produced by *K. intermedius*. *D. bruxellensis* was added simultaneously at concentrations of 10^6^ CFU/mL (BY_H0_) and 10^3^ CFU/mL (BY_L0_) and sequentially at 10^6^ CFU/mL (BY_H3_) and 10^3^ CFU/mL (BY_L3_). Means with different letters are significantly different (*p* < 0.05).

**Table 1 jof-08-01206-t001:** BC production experimental set up with *K. intermedius* mono-(B_-control_) and co-cultures with *D. bruxellensis* at 30 °C. *D. bruxellensis* was added simultaneously at concentrations of 10^6^ CFU/mL (BY_H0_) and 10^3^ CFU/mL (BY_L0_) and sequentially at 10^6^ CFU/mL (BY_H3_) and 10^3^ CFU/mL (BY_L3_).

KERRYPNX	Cell Concentration (log CFU/mL)		
Sample	*K. intermedius*	*D. bruxellensis*	Day of Yeast Addition
B_-control_	6	n.a.	n.a.
Y_-control_	n.a.	6	n.a.
BY_H0_	6	6	0
BY_L0_	6	3	0
BY_H3_	6	6	3
BY_L3_	6	3	3

## Data Availability

Not applicable.

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
