# Peer review of "The Effect of Dekkera bruxellensis Concentration and Inoculation Time on Biochemical Changes and Cellulose Biosynthesis by Komagataeibacter intermedius"

_jof, 2022, doi:10.3390/jof8111206_

Round 1

Reviewer 1 Report

The manuscript entitled "The Effect of Dekkera bruxellensis Concentration and Inoculation time on Bacterial Cellulose Synthesis by Komagataeibacter intermedius” is about co-cultures of bacteria and yeast to produce bacterial cellulose. The authors evaluated the effect of two different concentrations of yeast cells and two times of inoculation on the co-culture. The control cultures of yeast and bacteria in monoculture were studied. The importance of the work resides in the description of the changes in microbial growth, remaining sugars, BC yield, and some metabolites such as ethanol, carboxylic acids, and nitrogen. However, the biochemical dynamics and interactions of the cultures need to be analyzed in depth to explain the behavior. Finally, nowadays is not a novelty that cell concentration and inoculation time have crucial impacts on species interactions in the co-culture systems.

Some specific comments regarding the research report are shown below:

Line 11: I consider that “survival rates” is not the appropriate term because the study was focused on the “specific growth rate” and metabolites produced in the co-cultures.  

Section 2.2.2. (Line 96): I suggest to present the name of each treatment as a list or a table, to explain better the differences among all treatments.

Section 2.2.3. (Line 109): The samples from the culture were serially diluted and plated to count the CFU. There were used an antibiotic or antifungal to differentiate between yeast and bacteria? The PDA agar supplemented with NaCl and HS agar supplemented with acetic acid are not completely selective media.

Section 3.2. (Line 204): In this section, it is not properly discussed the reason for pH increasing. In general, this kind of fermentation produces a decrease in pH; the possible reasons for to reach a pH up to 9.0 are not presented.

Reviewer 2 Report

This study deals with the very interesting topic of bacterial cellulose production in co-culture systems inoculated with yeast and bacteria. The results showed that the inoculation rate of D. bruxellensis was very important to improve the BC production of K. intermedius. The article is overall well written and structured. However, I am not very clear on the methods used to adjust the concentration of microbial cells and evaluate the BC yield. In particular, the BC yield value in this study was much lower than previously reported data (Devanthi, P.V.P, et al. J. Fungi 2021, 7, 705, doi:10.3390/jof7090705). I think it is better to address the reasons for the difference of BC yield from various works in the discussion and to make these points clearer for readers.

Some comments are included below

Line 93, please describe the methods used to adjust the cell concentrations. It seems that the yeast concentrations at day 0 in Figure 2 are lower than adjusted goals; please explain why in the discussion.

Line 99, please add the inoculation rate for Y – control, is it about 106 CFU/mL?

Line 104, please change “1” as “14”?

Line 114, h-1 should be superscript?

Line 125, “no more pellet was observed”, was this visually assessed?

Line 141, what was the BC harvesting method?

Line 167-171, these have been shown in the section 2.2.2, I think there is no need to repeat here.

Line 233, please change “were observed” as “was detected”.

Line 236, is it day 5 or day 7?

Line 251, The gluconic acid concentrations at the 14th days are not distinguishable in Figure 4. It is better to magnify the data as an insert figure.

Line 287, Are there any ethanol data of Y – control?

Line 297, what is the meaning of “gain more advantage”? Please change the words if possible.

Line 378, “gluconic acid” means the peak concentration or the final. Since the final values are not distinguishable in Figure 4, please verify the results.

Line 394, please change “Figure 11” as “Figure 9”.

Reviewer 3 Report

The paper presented for review is written in a reasonably accessible language and in general it presents quite simple and basic research.

The introduction, methods as well as the presentation of results also, in general, does not raise any objections. The phrases "B-control was found", etc.,  sound a bit awkward and definitely need improvement.

Also, the title itself and the abstract should reflect the content of the work more, because reading them assumes that  BC biosynthesis is the main issue, while the authors presented a number  of biochemical parameters, dedicating the same amount of space to all  of them.
